# Study protocol: a multicentre, open-label, parallel-group, phase 2, randomised controlled trial of autologous macrophage therapy for liver cirrhosis (MATCH)

Paul Noel Brennan [ID],[1] Mark MacMillan,[1] Thomas Manship,[2] Francesca Moroni,[3] Alison Glover,[4] Catriona Graham,[5] Scott Semple,[6] David M Morris,[6] Alasdair R Fraser,[7] Chloe Pass,[7] Neil W A McGowan,[7] Marc L Turner,[7] Neil Lachlan,[8] John F Dillon,[9] John D M Campbell,[7] Jonathan Andrew Fallowfield,[10] Stuart J Forbes[11]

For numbered affiliations see end of article.

**Correspondence to**
Dr Paul Noel Brennan;
pbrenna2@ed.ac.uk

## ABSTRACT

**Introduction** Liver cirrhosis is a growing global healthcare challenge. Cirrhosis is characterised by severe liver fibrosis, organ dysfunction and complications related to portal hypertension. There are no licensed antifibrotic or proregenerative medicines and liver transplantation is a scarce resource. Hepatic macrophages can promote both liver fibrogenesis and fibrosis regression. The safety and feasibility of peripheral infusion of ex vivo matured autologous monocyte-derived macrophages in patients with compensated cirrhosis has been demonstrated.

**Methods and analysis** The efficacy of autologous macrophage therapy, compared with standard medical care, will be investigated in a cohort of adult patients with compensated cirrhosis in a multicentre, open-label, parallel-group, phase 2, randomised controlled trial. The primary outcome is the change in Model for End-Stage Liver Disease score at 90 days. The trial will provide the first high-quality examination of the efficacy of autologous macrophage therapy in improving liver function, non-invasive fibrosis markers and other clinical outcomes in patients with compensated cirrhosis.

**Ethics and dissemination** The trial will be conducted according to the ethical principles of the Declaration of Helsinki 2013 and has been approved by Scotland A Research Ethics Committee (reference 15/SS/0121), National Health Service Lothian Research and Development department and the Medicine and Health Care Regulatory Agency-UK. Final results will be presented in peer-reviewed journals and at relevant conferences.

**Trial registration numbers** ISRCTN10368050 and EudraCT; reference 2015-000963-15

## INTRODUCTION

Liver disease is responsible for almost 2 million deaths per year globally, 1 million directly relating to complications of end-stage liver failure and a further 1 million due complications of hepatitis including

## Strengths and limitations of this study

► First randomised controlled trial of an innovative cell-based therapy for cirrhosis.
► Range of evidence-based non-invasive assessments of liver fibrosis and function.
► Concurrent longitudinal measurement of health-related quality of life in an important chronic liver disease population.
► Open-label design, but outcome assessors blinded to treatment allocation.

hepatocellular carcinoma (HCC).[1] Cirrhosis and liver cancer are now, respectively, the 11th and 16th most common cause of death globally, accounting for 3.5% of all deaths. Variation in liver disease epidemiology occurs relative to the prevalence of modifiable risk factors including harmful alcohol ingestion, obesity/metabolic syndrome and viral hepatitis.[2] Worldwide there were 10·6 million prevalent cases of decompensated cirrhosis and 112 million prevalent cases of compensated cirrhosis in 2017.[3]

Cirrhosis represents the end-stage of chronic liver injury and progressive fibrosis (scarring), irrespective of the underlying aetiology. It is characterised by severe liver fibrosis leading to architectural disruption, hepatocyte dysfunction and portal hypertension. Cirrhosis typically affects those of working age, which has broad socioeconomic impacts. Furthermore, cirrhosis impairs health-related quality of life (HRQoL) including mental health and physical factors and reduced ability to perform activities of

daily living[4]; those with primary biliary cholangitis (PBC), non-alcoholic fatty liver disease (NAFLD) and hepatitis C virus (HCV) appear disproportionately affected.[5]

The classical dichotomy of chronic liver disease staging is compensated (asymptomatic) or decompensated cirrhosis. Acute decompensation delineates the development of one or more associated sequelae and is a key prognostic inflection point. The transition from compensated to decompensated cirrhosis occurs at a rate of about 5%–7% per year.[6] Decompensation represents a prognostic milestone as it significantly alters mortality, with a cumulative 1-year mortality of 77% for those with stage 3 and 4 decompensated disease vs 4.4% in those with compensated disease. Importantly, emergency hospitalisation for decompensated liver disease heralds a deterioration in a patient's prognosis independent of stage of cirrhosis.[7]

Cirrhosis decompensation heralds the development of widespread organ dysregulation, including portal hypertension, splanchnic vasodilation, left ventricular impairment and systemic immune dysfunction. Inflammatory mediators of liver disease may underpin and potentiate nitric oxide-mediated capillary dysfunction, direct immunocytopathy and induce significant metabolic derangement and redistribution of essential nutrient precursors.[8]

For patients in whom disease-specific therapy is unsuccessful or not possible, treatment options remain limited. Presently, although numerous agents have been evaluated in clinical trials, there are no approved pharmacological therapies for reversing fibrosis or stimulating liver regeneration in the cirrhotic liver.[9] Liver transplantation remains the only curative option for those with end-stage cirrhosis or HCC. Unfortunately, a significant proportion of those referred for transplant assessment are ineligible and ~12% die annually while on the waiting list in the UK.[10 11] Those who do undergo liver transplantation require lifelong immunosuppression with inherent risks of toxicity and adverse effects.[12]

Although whole organ or split liver transplantation are well established procedures to reinstate liver functional capacity, cell-based transplantation approaches are emerging.[13] Successful cell therapy could theoretically overcome organ availability limitations, while avoiding invasive surgical interventions. Successful hepatocyte transplantation involves reconstitution of as little as 1%–2.5% of functional tissue across a range of inherited metabolic liver diseases and highlights the utility of such approaches.[14] Furthermore, there is a requirement for treatments that can 'bridge' patients with cirrhosis until a donor organ is available or allow spontaneous regeneration to occur following acute liver failure. Cell therapies that sufficiently modulate cirrhosis by reducing fibrosis and stimulating liver function may also promote endogenous tissue repair and regeneration such that the need for transplantation is delayed or obviated.

Previous studies have typically focused on the use of mesenchymal stem cells (MSCs), hepatocyte stem cells and heterogeneous cell populations which will include proinflammatory and profibrotic cell lineages. Despite promising preclinical studies, randomised controlled trials (RCTs) of autologous cell therapies in cirrhosis have so far been disappointing.[15 16]

Macrophages are a heterogeneous, highly plastic population of cells with a diverse spectrum of roles within the liver including phagocytosis and maintenance of immune tolerance. Hepatic monocyte-derived macrophages are known to play a dual role in liver fibrosis. During chronic liver injury models they mediate the recruitment of proinflammatory cells and activation of hepatic stellate cells to promote fibrogenesis.[14] Conversely, fibrosis regression is characterised by an in situ phenotypic switch to a restorative hepatic macrophage population with pro-resolution properties[17] whereby liver repair and regeneration is facilitated by increased expression of matrix metalloproteinases (MMPs), growth factors and phagocytosis-related genes.[18 19] This process of phenotypic 'switching' from a proinflammatory 'M1-like' moiety, to a pro-resolution 'M2-like' macrophage is mediated via down-regulation of NOD-containing, LRR-containing and pyrin domain-containing protein 3.[14]

In a mouse model of chronic liver injury, cell therapy with unmanipulated syngenic macrophages reduced fibrosis and improved markers of liver function.[20] Furthermore, infusion of human macrophages (differentiated from cirrhotic patients' apheresis-derived CD14+ monocytes) also resolved liver fibrosis in mice, indicating their suitability for clinical therapy.[18 20 21]

We recently demonstrated the feasibility of performing apheresis in cirrhotic patients and differentiating autologous bone marrow derived monocytes into macrophages.[22] This process includes specific CD14+ monocyte isolation from peripheral circulation leucopharesis collections using CliniMACS automated separation device, a closed system, where the product is incubated with CD14 labelled magnetic beads, allowing separation of CD14+ cells when passed over a magnetic column. Selected CD14+ monocytes are counted and resuspended in differentiation medium containing 100 ng/mL macrophage colony-stimulating factor (M-CSF). Cells are placed into closed system, low adhesion culture bags at optimum cell density ($2 \times 10^6$ cells per mL and per $cm^3$). Cells are cultured in a humidified atmosphere at 37°C, with 5% CO2, for 7 days. Media replenishment is undertaken twice during culture (typically days 3 and 5), using differentiation media supplemented with 100 ng/mL M-CSF. Flow cytometry is used to determine cell viability and phenotype cell populations premonocyte and postmonocyte selection and postmacrophage differentiation prior to product release, this has been validated for 7-day and 10-day time points.

We also have extensive preclinical data demonstrating that peripherally injected macrophages hone to the liver (predominantly) and spleen (after passing rapidly through the lungs) and that this process in enhanced in the presence of liver damage.[20 23] Furthermore, in a first-in-human study we confirmed the safety, feasibility

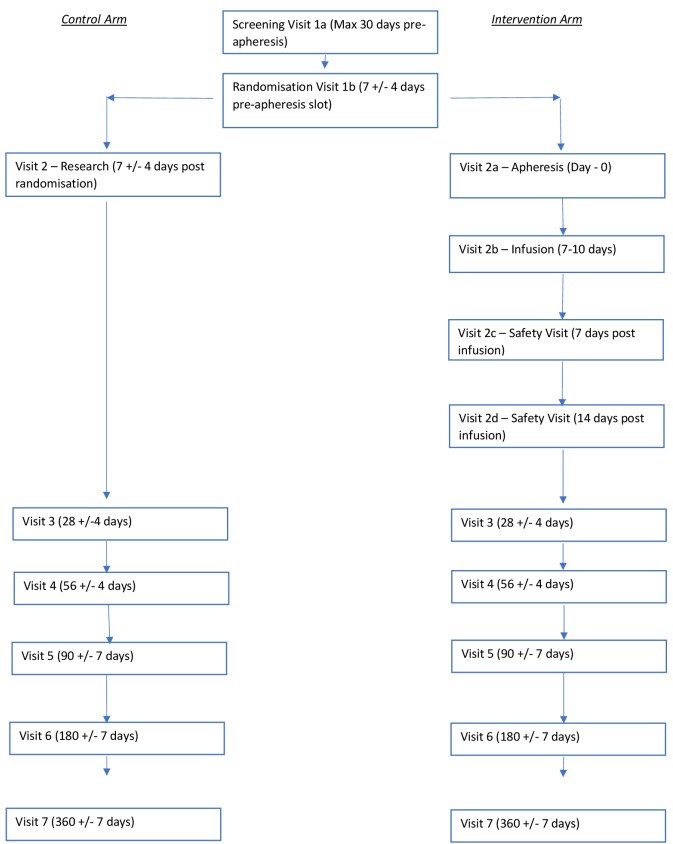

**Figure 1** Schematic of trial timeline.

and maximum achievable dose of autologous macrophages.[24] The study was not controlled, and therefore, unable to evaluate efficacy. However, we observed some initial signals related to enhanced fibrosis remodelling and liver function that warranted assessment in an RCT as presented here.

## Objectives

The primary objective of this phase 2 RCT is to evaluate whether there is an improvement in liver function at 3 months in patients receiving autologous macrophage therapy compared with standard medical care.

The secondary objectives are to assess any improvement in markers of liver fibrosis, increased disease related quality of life, reduced liver related clinical events and prolonged transplant-free survival.

## Trial design

The MATCH trial is designed as a multicentre, open-label, parallel-group, phase 2, RCT to compare autologous macrophage therapy with standard medical care in patients with compensated cirrhosis. Randomisation will be performed with a 1:1 allocation ratio and the primary outcome is the baseline to 90-day change in Model for End-Stage Liver Disease (MELD) score. Figure 1 provides an overview of trial pathway following randomisation to the respective arms. Initially, the proposed trial was designed to administer three infusions to those randomised to the treatment arm. It became apparent that it would not be acceptable or feasible to continue

with three infusions due to the onerous commitment required of participants and the challenge to complete the trial within the proposed time frame. Therefore, as a pragmatic approach, and in line with the phase 1 study, it was decided that a single infusion protocol should be adopted to simplify the participant journey and ensure adequate recruitment. This was agreed with the trial steering committee (TSC), sponsor and data monitoring committee (DMC).

## METHODS
### Study oversight

The MATCH 0.1 trial is an investigator-led study, funded by the Medical Research Council (reference MR/M007588/1) and sponsored by ACCORD (Academic and Clinical Central Office for Research and Development for National Health Service (NHS) Lothian/University of Edinburgh). Trial oversight is also provided by a TSC and DMC, who are impartial around aspects of study design and logistics but provide independent advice and interval safety analyses. The study started initially in 2016 and is likely to continue until late 2022. All study-related documents were designed by the trial team with input from ACCORD, an independent statistician and the Scottish National Blood Transfusion Service (SNBTS) team. The trial will be conducted according to the ethical principles of the Declaration of Helsinki 2013 and has been approved by Scotland A Research Ethics Committee (reference 15/SS/0121), NHS Lothian Research and Development department and the Medicine and Health Care Regulatory Agency (MHRA-UK). Good Clinical Practice regulations will be followed and written informed consent will be obtained from all participants.

### Study setting

The MAcrophage Therapy for iver CirrHosis (MATCH) trial is recruiting in three hepatology centres in Scotland: Royal Infirmary of Edinburgh (Tertiary Transplant Centre/level 3 hepatology services), Ninewells Hospital, Dundee and Glasgow Royal Infirmary (both level 2 hepatology centres). There are plans to potentially extend recruitment to include additional sites.

### Patient and public involvement

There was no direct patient or public involvement groups involved in the study design. The overall study design was developed from previous experience of the investigators involved in the design and coordination of similar studies.

## ELIGIBILITY CRITERIA (INCLUSIONS/EXCLUSIONS)
### Inclusion criteria

► Aged between 18 and 75 years (inclusive) at time of screening
► Aetiology: One or more of:
  – Alcohol-related liver disease (no active alcohol misuse ≥6 calendar months prior to screening).

Features of chronic liver disease with a compatible history of alcohol excess (>80 g/day), in the absence of other causes of chronic liver disease.

- PBC 2 out of: Cholestatic liver function tests (LFTs), Positive antimitochondrial antibody (titre >1:40). Compatible liver histology (if already receiving ursodeoxycholic acid must be established on current dose >3 months prior to enrolment).
- NAFLD Either: Histological evidence of hepatic steatosis in the absence of other liver diseases Or: Imaging compatible with NAFLD (eg, fatty infiltration of liver) and one or more risk factors (eg, elevated body mass index, type 2 diabetes mellitus, hypertriglyceridaemia, hypertension) and the absence of significant alcohol consumption (<20 g/day) and no evidence of other causes of chronic liver disease.
- Cryptogenic cirrhosis diagnosis of cirrhosis unattributable to any other cause.
- Haemochromatosis diagnosis made on basis of compatible biochemistry (transferrin saturation >60%, ferritin >400), genotype (homozygous C282Y or H63D compound heterozygote) or histology.
- Alpha-1 antitrypsin deficiency diagnosis based on compatible genetic, phenotypic or histological testing.
- Previous chronic hepatitis C (sustained viral response ie, undetectable HCV RNA 24 weeks after treatment).

► Diagnosis of cirrhosis—invasive or non-invasive criteria cirrhosis defined as any of:
- Biopsy-confirmed diagnosis of cirrhosis.
- Transient elastography (TE)—≥15 kPa.
- Clinical and radiological features which in the opinion of the investigator correlate with a diagnosis of cirrhosis.

► A MELD score (Pre-2016) of ≥10 and ≤17 at screening visit.

### Exclusion criteria

Refusal or inability to give written informed consent to participate in the study.

► Other causes of chronic liver disease/cirrhosis not included in the listed aetiologies
► Portal hypertensive haemorrhage; active episode of bleeding requiring hospitalisation in the last 3 months where varices have not been eradicated by endoscopic band ligation or transjugular intrahepatic portosystemic shunts (TIPSS).
► Ascites unless, in the opinion of the investigator, is minimal and well controlled with no increase to diuretic therapy in the last 3 months.
► Hepatic encephalopathy; current or requiring hospitalisation for treatment in the last 3 months.
► HCC—uncertain cases to be discussed at the local hepatobiliary multidisciplinary team meeting. Dysplastic or indeterminate nodules to be excluded;

regenerative or other nodules to be included at discretion of investigator.
► Previous diagnosis of HCC.
► Previous organ transplant recipient.
► Listed for liver transplantation.
► Any situation that in the investigators opinion may interfere with optimal study participation such as alcohol or drug abuse, domicile too distant from study site, potential non-compliance or inability to cooperate.
► Presence of clinically relevant acute illness which may preclude on basis of safety.
► Presence or history of cancer with exception of adequately treated localised skin carcinoma, in situ cervical cancer or solid malignancy excised in total, with no recurrence (5-year interval).
► Pregnancy or breast feeding.

### Interventions

Participants who are randomised to the treatment arm will receive an infusion of the maximum achieved dose up to $1 \times 10^9$ (day 0). The apheresis product will be collected under the terms of the Human Tissue (Quality and Safety for Human Application) Regulations 2007 No. 1523 enacting the requirements of the EU Tissues and cells Directive (2004/2023) and associated Commission Directives at the Apheresis Unit (Royal Infirmary of Edinburgh, Edinburgh, UK). CD14+ monocytes will be isolated, and the macrophage cell product will be manufactured as previously described,[25] in compliance with GMP regulations under the terms of the SNBTS MIA (IMP) licence at the SNBTS Cell Therapy Facility (Scottish Centre for Regenerative Medicine, Edinburgh, UK).

Each patient will be monitored closely during the infusion to identify potential hypersensitivity reactions and 4 hours postinfusion bloods to monitor for any evidence of macrophage activation syndrom. A total of 28 participants will be randomised to standard medical care and 28 to receive the cell infusion, allowing for original estimate of 5 dropouts from each arm. Additional safety data will be collected for the first infusion only for the first three patients randomised to the treatment arm. If it has not been possible to achieve $1 \times 10^9$ macrophages, then the participants will be infused with the quantity obtained, with minimum concentration being $1.25 \times 10^8$ cells. This minimum cell concentration was derived from previous validation work and is stipulated as part of the product release criteria as designated by the MHRA.

### Outcomes
#### Primary outcome measure
##### Model of End-Stage Liver Disease
The MELD was originally devised to predict survival in patients with complications of portal hypertension undergoing elective placement of TIPSS. The algorithm is based on: creatinine, bilirubin and prothrombin ratio and has been demonstrated to be superior to the Child-Turcotte-Pugh score in predicting 3-month mortality

among patients with end-stage liver disease.[26] However, the MELD score has also been applied to predict survival in patients with cirrhosis with infections, variceal haemorrhage, and those with fulminant hepatic failure and alcoholic hepatitis.[27]

## Secondary outcome measures
### Transplant-free interval

The number of participants in each of the two treatment arms who are transplant free at 12 months will be expressed as proportions and a binomial test will be used for the comparison of proportions between the treatment arm and the control arm. The difference in proportions will be presented along with the 95% CI for the difference in the proportions.

The time to death or transplant will be presented using a Kaplan-Meier survival curve stratified by treatment and accompanied by a log-rank statistic comparing the two arms. Survival estimates with be presented by treatment arm at 3, 6, 9 and 12 months.

### Non-invasive markers of fibrosis

Changes in our secondary outcome measures over 90 days up to maximal 360 days as per schedule (table 1), these include: serum enhanced liver fibrosis (ELF) test (iQur, London, UK, serum Protein Fingerprint markers (Nordic Bioscience, Herlev, Denmark), hepatic TE (Echosens, Paris, France) and the UK End-Stage Liver Disease (UKELD) score.

### Enhanced liver fibrosis

A standardised clinically validated immunoassay test measuring three serum biomarkers which have been shown to correlate to the level of liver fibrosis assessed by liver biopsy, comprising:
► Hyaluronic acid.
► Tissue Inhibitor of Metalloproteinase 1.
► Aminoterminal propeptide of type III procollagen.

The concentrations of each individual protein marker are combined in an algorithm which produces a composite score related to the level of liver fibrosis. The ELF score is a sensitive, specific and validated method for the non-invasive assessment of hepatic fibrosis in mixed, HCV and NAFLD patient groups.[28]

### Protein Fingerprint biomarkers

During extracellular matrix (ECM) turnover, proteolytically cleaved matrix degradation fragments or neoepitopes, are released into the systemic circulation. Cleavage of each ECM protein by specific MMPs generates a unique neoepitope. These neoepitopes are more accurate diagnostic and prognostic markers for individual fibroproliferative diseases than their protein of origin. These novel serum biomarkers have been shown to identify patients with progressive fibrosis and permit monitoring of the response to antifibrotic therapy,[29] and also correlate with portal hypertension in patients with cirrhosis.[30]

### TE (Fibroscan)

TE is a non-invasive method for assessing liver fibrosis. Mild amplitude and low frequency vibrations (50 Hz) are transmitted to the liver tissue, inducing an elastic shear wave that propagates through the underlying liver tissue. The velocity of the wave is directly related to tissue stiffness, considered as a surrogate of the amount of fibrotic tissue. This is expressed as a numerical value in kilopascals (kPa). It is reliable, reproducible with high intraobserver and interobserver agreement and has been validated in most causes of chronic liver disease[31]

### Chronic Liver Disease Quality of Life Questionnaire

The Chronic Liver Disease Quality of Life Questionnaire (CLDQ) is a liver-specific questionnaire for measuring HRQoL in participants with chronic liver disease. It is self-administered, takes approximately 10 min to complete and is designed to reflect the 2 weeks prior to testing. If necessary, participants can request help to complete this.[32]

It includes 29 items divided into 6 quality of life domains: Abdominal symptoms, Fatigue, Systemic symptoms, Activity, Emotional function and Worry. These items are ranked on a 1–7 scale, providing a possible range of scores from 29 (worst quality of life) to 203 (best quality of life). The construct validity of the CLDQ was supported by a strong correlation with participant's global rating scores. It has been shown to be valid and has good test–retest reliability.[33–35]

### UKELD score

The UKELD score is readily performed incorporating routine biochemical and haematological indices including bilirubin, albumin, alanine transaminase (ALT) and International Normalised Ratio (INR). The UKELD score was developed by the UK Liver Transplant Units to predict transplant waiting list mortality.[36]

The score uses the parameters of Bilirubin (Bil), INR, Creatinine (Creat) and Sodium (Na) in the following algorithm:

$$UKELD = [(5.395*ln(INR)) + (1.485*ln(Creat) + (3.130*ln(Bil)) - (81.565*ln(Na))] + 435$$

### MRI and MR spectroscopy

MRI and spectroscopy (MRS) provide methods for the non-invasive assessment of liver microstructure and function. MRI allows for imaging biomarkers to be determined using Liver*Multi*Scan.[37] Tissue microstructure will be investigated using clinically validated metrics. Fibrosis will be assessed by $cT_1$, iron content with $T_2^*$ and the amount of fat in the liver using proton density fat fraction. Organic phosphorus in the liver can be quantified with Phosphorus-31 ($^{31}P$) MRS[38] a more explorative technique. Using $^{31}P$ MRS energy metabolism may be investigated via ATP levels and cell membrane integrity by measuring precursors and degradation products. The paired imaging of this study allows for the current utility of MRI to assess disease progression and treatment response to be evaluated

**Table 1** Trial assessment schedule

| | | | Treatment group | | | | Control group | | | | | |
|---|---|---|---|---|---|---|---|---|---|---|---|---|
| | Visit 1a | Visit 1b | Visit 2a Within 7±4 days of Visit 1b | Visit 2b 7>10 days after apheresis (Day 0) | Visit 2c (Day 7) | Visit 2d (Day 14) | Visit 2 (day 7±4 days from visit 1b) | Visit 3 (Day 28±4 days) | Visit 4 (Day 56±4 days) | Visit 5 (Day 90±7 days) | Visit 6 (Day 180±7 days) | Visit 7 (Day 360±7 days) |
| Informed consent | X | | | | | | | | | | | |
| Clinical assessment | X | X | X | X | X | X | X | X | X | X | X | X |
| Vital signs | X | X | X | X | X | X | X | X | X | X | X | X |
| Screening blood tests | X | | | | | | | | | | | |
| ECG | X | | | | | | | | | | | |
| Standard blood tests | X | X | | X | X | X | X | X | X | X | X | X |
| Research bloods* | X | X | X | | X | X | X | X | X | X | X | X |
| Mandatory microbiology | X | | X | | | | | | | | | |
| Ferritin | X | | | X† | X | X | | | | | | |
| Triglyceride | X | | | X† | X | X | | | | | | |
| Pre-infusion blood tests | | | | X | | | | | | | | |
| MELD/UKELD | X | X | X | X | X | X | X | X | X | X | X | X |
| Pregnancy test | X‡ | X‡ | | X‡ | X§ | | | | | | | X‡ |
| Abdominal Ultrasound Scan (USS) | X¶ | | | | | | | | | | X¶ | X¶ |
| Fibroscan | X | | | | | | | X | | X | X | X |
| ELF panel | X | | | | | | | X | X | X | X | X |
| Protein fingerprint | X¶ | | | | | | | X¶ | | X¶ | | |
| CLDQ | | X | | | | | | | | X | X | X |
| 31P MRS MRI** | X* | | | | | | | | | | | |
| Adverse events | X | X | X | X | X | X | X | X | X | X | X | X |
| Clinical events | X | X | X | X | X | X | X | X | X | X | X | X |
| Concomitant Medication | X | X | X | X | X | X | X | X | X | X | X | X |

*If pass screen and before visit 2b.
†Obtain before discharge.
‡women of child bearing age only.
§If test not carried out at previous visit.
¶Fasted visit.
**Royal Infirmary of Edinburgh (RIE) patients only.
CLDQ, Chronic Liver Disease Quality of Life Questionnaire; ELF, enhanced liver fibrosis; MELD, Model for End-Stage Liver Disease; MRS, MR spectroscopy; UKELD, UK Model for End-Stage Liver Disease.

MRI data collected is exploratory and will be according to subgroup analysis: the only planned subgroup analysis is to present the primary outcome for the RCT by disease aetiology (ALD, NAFLD, other). MRI is performed at index visit 2 (or within 7 days) and again at primary outcome time point of 90 days (±7).

## Sample size/power calculation

To detect a difference in the baseline to 90-day change in MELD score of 1 SD using a two-sided, two-sample test with a 5% level of significance, a sample size of 23 per group to detect the same level of difference with 90% power is required. All analyses will be carried out on an intention to treat basis, retaining participants in their randomised treatment groups irrespective of the treatment received. Adverse event (AE) data will be presented by treatment received.

The number of participants who do not adhere to the protocol is expected to be low. All protocol violations and ineligible participants will be recorded.

## Recruitment

### Identification of potential patients

Potential participants will be identified by their usual direct healthcare team. The treating physician will either introduce the individual to the trial team or ask permission for the trial team to contact them; this could be done through a dedicated invitation letter or a telephone call. The participant information sheet will be provided and there will be an opportunity to ask questions. If they agree, a further visit will be scheduled to discuss trial enrolment. This will take place no less than 24 hours later.

### Randomisation

Following confirmation of the participant meeting the eligibility criteria, a delegated member of the research team will enter minimal information (participant id, and aetiology) into an online randomisation system, produced for the study by Edinburgh Clinical Trials Unit to determine the treatment allocation. At randomisation, patients will be allocated a unique patient trial number and scheduled for treatment and follow-up visits as detailed in the trial schedule.

### Allocation

Participants will be assigned to receive either standard medical care or to receive a fresh dose of autologous MDMs at the maximum achievable dose, in a 1:1 ratio based on a minimisation algorithm using the key variable aetiology of disease (ALD, NAFLD, other.) To ensure the allocation is random, participants will be assigned to the group which minimises the imbalance with probability 0.8. If a participant falls into two or more strata, then the dominant aetiology (as determined by treating physician) will be used.

### Blinding

Due to the nature of the intervention neither participants nor staff can be blinded to allocation of treatment.

For some of the additional secondary outcomes we will maintain blinding of external assessors including those processing samples for ELF and protein fingerprint markers. Similarly, there is blinding of MRI physicists and external validation companies responsible for experimental MRI interpretation.

## Data collection

The case report form (CRF) will be completed at set time points as per trial schedule. The CRF will be completed by the investigator or an authorised member of the research team (as delegated on the Site Signature and Delegation Log). The exception is the serious AE Form which must be signed by the investigator.

Data reported in each form should be consistent with the source data or the discrepancies should be explained. If information is not known, this must be clearly indicated in the form.

Completed CRFs submitted to the clinical research facility will be reviewed by the trial coordinator. The data will be entered into an electronic database by designated members of the trial team.

## Data management

The following personal data will be collected as part of the research: name, date of birth and CHI numbers (Community Health Index; a unique is a 10-character numeric identifier, allocated to each patient on first registration with the NHS system in Scotland). Personal data will be stored in locked cabinets by the research team at the clinical research facilities at each site. Personal data will be stored for 30 years in keeping with the blood safety and quality regulations. The University of Edinburgh and NHS Lothian are joint data controllers along with any other entities involved in delivering the study that may be a data controller in accordance with applicable laws.

All investigators and study site staff involved with this study must comply with the requirements of the appropriate data protection legislation (including where applicable the general data protection regulation regarding the collection, storage, processing and disclosure of personal information. Access to personal information will be restricted to individuals from the research team treating the participants, representatives of the sponsor(s) and representatives of regulatory authorities.

Study data will be collected and managed using Research Electronic Data Capture (REDCap) electronic data capture tools hosted at The University of Edinburgh. REDCap[39] is a secure, web-based application designed to support data capture for research studies, providing: an intuitive interface for validated data entry; audit trails for tracking data manipulation and export procedures; automated export procedures for seamless data downloads to common statistical packages; and procedures for importing data from external sources.

Published results will not contain any personal data that could allow identification of individual participants.

## Statistical analysis plan

The baseline to 90-day change in MELD score will be compared in the two treatment arms using a two-sample t-test or non-parametric equivalent as appropriate. MELD scores calculated for each participant throughout the trial will be used to calculate an area under the curve (AUC) and this will be compared across the groups using a two-sample t-test or non-parametric equivalent as appropriate. In the event of varying durations in the trial follow-up, the average AUC per month will be used so that all participants have a comparable measurement.

Changes in secondary outcome measures (ELF score liver stiffness, CLDQ score, transplant-free survival, number of clinical events, UKELD score, blood parameters (bilirubin, albumin, ALT, INR)) over the 1-year study period will be presented graphically by dose. Similarly, these results will used to calculate an AUC for each participant and will be compared across the groups using a two-sample t-test or non-parametric equivalent as appropriate.

The only planned subgroup analysis is to present the primary outcome by disease aetiology (ALD, NAFLD, other). Primary data analysis will be conducted on participants who receive a single infusion versus control; the primary analysis will then be repeated to include those subjects who receive more than one infusion (three individuals). There are no plans for an interim analysis.

## Data monitoring

The trial will be coordinated by a project management group, consisting of the grant holders (chief investigator and principal investigator in Edinburgh), a trial manager and coordinating nurse.

The trial manager will oversee the study and will be accountable to the chief investigator. The trial manager, or an authorised member of the research team, will be responsible for checking the CRFs for completeness, plausibility and consistency. Any queries will be resolved by the Investigator or delegated member of the trial team. A Delegation log will be prepared detailing the responsibilities of each member of staff working on the trial.

## Safety assessments

The investigator is responsible for the detection and documentation of events meeting the criteria and definitions detailed within the protocol (available on request). Full details of contraindications and side effects that have been reported following administration of the IMP can be found in the relevant investigator's brochure.

Participants will be instructed to contact their Investigator at any time after consenting to join the trial if any symptoms develop. All AEs that occur after joining the trial must be reported in detail in the CRF or AE form. In the case of an AE, the investigator should initiate the appropriate treatment according to their medical judgement. Any AE events still present on day 360 will be confirmed and recorded as 'ongoing' in the CRF. If appropriate, these should be handed over to the participants' general practitioner or direct care team.

The ACCORD Research Governance and QA Office is responsible for pharmacovigilance reporting on behalf of the cosponsors (University of Edinburgh and NHS Lothian).

The ACCORD Research Governance and QA Office has a legal responsibility to notify the regulatory competent authority and relevant ethics committee (Research Ethics Committee (REC) that approved the trial). Fatal or life threatening Suspected Unexpected Serious Adverse Reactions (SUSARs) will be reported no later than seven calendar days and all other SUSARs will be reported no later than 15 calendar days after ACCORD is first aware of the reaction.

ACCORD will inform investigators at participating sites of all SUSARs and any other arising safety information.

An annual safety report/development safety update Report will be submitted, by ACCORD, to the regulatory authorities and RECs listing all SARs and SUSARs.

## Monitoring and oversight

An ACCORD clinical trials monitor, or an appointed monitor will visit the investigator site prior to the start of the study and during the course of the study if required, in accordance with the monitoring plan if required. Risk assessment will determine if audit, by the ACCORD QA group, is required. Details will be captured in an audit plan.

## DISCUSSION

MATCH is an RCT designed to identify whether there is a measurable improvement in MELD score and also in relevant secondary clinical outcomes, HRQoL and non-invasive biomarkers following autologous macrophage therapy. It builds on the safety and feasibility assessment of the earlier phase I trial. Recent Food and Drug Administration (FDA) guidance on development of treatments for cirrhosis has indicated there are no acceptable surrogate endpoints (eg, histological improvement) so our focus in this study is on clinically meaningful assessments such as liver function, survival and HRQoL rather than liver biopsy.

Previous clinical trials using MSCs across a range of aetiologies of liver disease have yielded mixed results. In trials which reported efficacy, the apparent benefit was transient, with no long-term improvement.[40 41]

One important rationale for using macrophages relates to the lack of efficacy of haematopoetic stem cells,[42] inherent challenges of using transplanted hepatocytes, and potential risk of introducing transplanted hepatocytes MSCs into a hostile host niche. Previous trials have demonstrated concerns around cellular engraftment and expansive potential of such approaches.

Preclinical studies undertaken by our group have administered macrophages via the portal vein, tail vein or intrasplenic route, but in our phase 1 trial we successfully used peripheral intravenous infusion which is safer and more convenient. While there is no cell-tracking

technique used in this trial to assess cell engraftment/durability, animal models and human case reports suggest that macrophages infused via either peripheral or central veins will transiently pass through the lungs, before engrafting in the liver and spleen.[21] However, hepatic artery or portal venous administration are considerably more invasive, with concerns regarding risk of bleeding and vessel injury,[43] and problems related to reversal of portal flow/porto-systemic shunting or splanchnic vessel thrombosis.[14]

Through this trial, we aim to add to the collective knowledge of this potential new therapeutic modality for liver disease in this patient population who currently have limited treatment options. If effective, autologous macrophage cell therapy will improve clinical outcomes and enhance HRQoL in people with cirrhosis.

Following initial trial results, we expect that a further extended study will be necessary to determine longer-term safety and the durability of treatment responses. Moreover, it is not yet clear whether patients may require repeat treatments to maximise efficacy.

We hope that this initial phase II trial will provide robust evidence to support and inform future trial design.

## ETHICS AND DISSEMINATION

The trial will be conducted according to the ethical principles of the Declaration of Helsinki 2013 and has been approved by Scotland A Research Ethics Committee (reference 15/SS/0121), NHS Lothian Research and Development department and the MHRA-UK. Good Clinical Practice regulations will be followed and written informed consent will be obtained from all participants. Results will be disseminated through peer-reviewed publications, presented at conferences and published on ClinicalTrials.gov. Ownership of the data arising from this study resides with the study team and their respective employers. The study team will follow the International Committee of Journal Editors guidelines. Requests for data access should be sent to the corresponding author (ORCID: 0000-0001-8368-1478).

**Author affiliations**
[1]Centre for Regenerative Medicine, The University of Edinburgh Medical School, Edinburgh, UK
[2]CLDD, NHS Lothian, Edinburgh, UK
[3]Department of Gastroenterology, NHS Grampian, Aberdeen, UK
[4]Scottish National Blood Transfusion Service, Edinburgh, UK
[5]Deanery of Clinical Sciences, The University of Edinburgh, Edinburgh, UK
[6]Centre for Cardiovascular Science, The University of Edinburgh Deanery of Clinical Sciences, Edinburgh, UK
[7]Tissues, Cells and Advanced Therapeutics, SNBTS, Edinburgh, UK
[8]Department of Gastroenterology, NHS Greater Glasgow and Clyde, Glasgow, UK
[9]Liver Group, University of Dundee Division of Cardiovascular and Diabetes Medicine, Dundee, UK
[10]Queen's Medical Research Institute, University of Edinburgh MRC Centre for Inflammation Research, Edinburgh, UK
[11]Centre for Regenerative Medicine, The University of Edinburgh Deanery of Clinical Sciences, Edinburgh, UK

**Contributors** FM and SJF was responsible for the conceptualisation and design of the trial. PNB is study clinician and drafted manuscript and provided critical review of protocol. JAF aided manuscript preparation and critical appraisal. CG was responsible for statistical design. AG, CP, NWAM ARF, MLT and JDMC were responsible reviewing sections around product manufacture. MM and TM provided manuscript review and critique. SIKS and DMM developed section on MRI imaging. NL and JFD provided critical appraisal of manuscript. All authors critically revised and approved the manuscript.

**Funding** This work was supported by a Medical Research Council UK grant (Biomedical Catalyst Major Awards Committee, Reference: MR/M007588/1) to SJF.

**Competing interests** PNB has received honoraria from Takeda. JAF has received consultancy fees for Ferring Pharmaceuticals, Macrophage Pharma, Aquilla BioMedical, Caldan Therapeutics, Cypralis, Third Rock Ventures, Rallybio, Narrow River Management, Gilde Healthcare, Guidepoint, Techspert.io and acted as advisory board member for: Novartis, Galecto Biotech, Tectonic Therapeutic and received research grant funding from Novartis and Intercept Pharmaceuticals. JFD has received honoraria and research grants from Gilead, AbbVie and MSD. JDMC and SJF are founders and scientific advisers to Resolution Therapeutics.

**Patient and public involvement** Patients and/or the public were not involved in the design, or conduct, or reporting, or dissemination plans of this research.

**Patient consent for publication** Not applicable.

**Provenance and peer review** Not commissioned; externally peer reviewed.

**ORCID iD**
Paul Noel Brennan http://orcid.org/0000-0001-8368-1478

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
