## [Reviewer comments · BMJ Open]

ARTICLE DETAILS

TITLE (PROVISIONAL)	Study Protocol: A multicentre, open-label, parallel-group, phase 2, randomised controlled trial of autologous macrophage therapy for liver cirrhosis (MATCH)
AUTHORS	Brennan, Paul; MacMillan, Mark; Manship, Thomas; Moroni, Francesca; Glover, Alison; Graham, Catriona; Semple, Scott; Morris, David; Fraser, Alasdair; Pass, Chloe; McGowan, Neil; Turner, Marc; Lachlan, Neil; Dillon, John; Campbell, John; Fallowfield, Jonathan; Forbes, Stuart

VERSION 1 – REVIEW

REVIEWER	Sharma, Mithun Department of Gastroenterology and Hepatology, Asian Institute of Gastroenterology
REVIEW RETURNED	12-Jul-2021

GENERAL COMMENTS	This is very important study which however after many years of use still remains a proof of concept The limitations which needs to be clearly discussed included in discusion 1. Using macrophage instead of autologous HSC or MSC - the logic behind this ?2. Why the route of administration was preferred3. Any plan of doing liver biopsy to prove homing in of cells
---

REVIEWER	Lachmann, Nico Hannover Medical School
REVIEW RETURNED	16-Jul-2021

GENERAL COMMENTS	Study Protocol: A multicentre, open-label, parallel-group, phase 2, randomised controlled trial of autologous macrophage therapy for liver cirrhosis (MATCH) This manuscript clearly demonstrates the study design and methodology, increasing its transparency and making it easier for others to observe and understand any alterations from the proposed protocol that might happen during the entire clinical trial. The authors openly discuss the problem (liver cirrhosis), highlighting the importance of new treatments, as of this moment there is the lack of a successful treatment other than organ transplantation. Overall, the authors did a great job, they included several key aspects such as study hypothesis, study design, eligibility, feasibility or schema/flowchart. Additionally, it is written in a simple
--

	and comprehensive way allowing to have a full view of the study protocol. However, there are some aspects to review:  1. Although the authors detailly explain the underlying disease background, the rational for using a macrophage therapy is rather short and only briefly refers to previous studies by the group (page 9, middle paragraph). Maybe this part could be a bit extended to further balance the introduction. 2. It is not clear to us, why the authors decided to change the trial design regarding the numbers of infusion (accepted by the trial steering committee): Are the authors expecting the same efficacy with one infusion in comparison to three? Have there be any preclinical studies comparing the efficiencies? Why have 3 patients been treated with 3 doses? And are those also the patients on which the additional safety analysis will be performed? 3. More from a scientific perspective: Do you think that through infusion, the majority of the macrophages will reach the target organ? Have you thought about a different cell delivery system in order to deliver macrophages directly to the damaged liver? 4. How was the minimum cell concentration of 1.25×10^8 defined? 5. In the discussion part, you mentioned that if the treatment is effective, autologous macrophage cell therapy could improve clinical outcomes and enhance HRQoL in people with cirrhosis. Don't you want to say that if effective, the therapy will improve the outcome? Otherwise, how can you determine if the treatment is effective if it might not improve the outcome? 6. Can the authors briefly explain how the macrophages are generated from patients' monocytes and not only refer to the reference? Did the authors (priviously) proof that the cells are the same whether infused 7 or 10 days post differentiation or what is the reason for that wide time range? Minor comments:  1. Page 9, line 18/19 still contains a comment "REF-use one of any reviews" instead of citing a review 2. Page 15, line 34/35: "as previously described24s": delete the "s" after the refence 3. Page 26, line 11/12. Please define SUSARs at the first mentioning
--	--

VERSION 1 – AUTHOR RESPONSE

Reviewer 1:

1. *Using macrophage instead of autologous HSC or MSC - the logic behind this?*

In the only adequately powered randomised controlled trial, we showed that haemopoietic stem-cell infusion was ineffective in patients with cirrhosis.¹ Clinical trials utilising mesenchymal stem cells have been completed for a variety of liver indications with mixed results. Given that recruited macrophages play a crucial role in orchestrating liver regeneration^{2,3} and the resolution of fibrosis^{4,5}, we hypothesised that exogenous ex vivo-differentiated macrophages with the appropriate phenotype could be administered to accelerate liver disease regression. In a mouse model of chronic liver injury, cell therapy with unmanipulated syngenic macrophages reduced fibrosis and improved markers of liver function^{6,7} (and infusion of human

monocyte derived macrophages have been shown to be anti-fibrotic in a mouse model of liver fibrosis⁷ Furthermore, in a first-in-human study⁸ we went on to confirm the safety, feasibility, and maximum achievable dose of autologous macrophages in cirrhotic patients. Within the data, initial signals suggested enhanced fibrosis remodelling and liver function that warranted assessment in a phase II randomised controlled trial. This is articulated in the paper (Page 7, lines 1-21).

2. Why the route of administration was preferred?

We have extensive pre-clinical data showing that peripherally injected macrophages home to the liver (predominantly) and spleen (after passing rapidly through the lungs) and that this process is enhanced in the presence of liver damage^{6,9}

This phase II study builds upon the preceding phase I study which was undertaken by our group using peripheral i.v. infusion of macrophages. Whilst primarily a safety and feasibility study, we observed signals suggesting a positive effect on fibrosis remodelling. Therefore, we chose to use the same route of administration here that was used to establish the maximum achievable dose. Furthermore, alternative approaches such as hepatic artery or portal vein administration are considerably more invasive, with safety concerns regarding risk of bleeding, vessel injury, and problems related to reversal of portal flow/porto-systemic shunting and splanchnic vessel thrombosis. Indeed, a study of cell therapy for liver cirrhosis reported a complication of hepatic artery dissection.¹⁰
(Page 8, lines 10-17)

3. Any plan of doing liver biopsy to prove homing in of cells

Our cells were not labelled and would not be detectable on biopsy (we have tested labelling with SPIOs in pre-clinical models but this changed the phenotype of the macrophages). Preclinical studies undertaken by our group have administered macrophages via the peripheral tail vein, portal vein or intrasplenic route, with assessment of cell engraftment. Whilst there is no cell-tracking technique used in this trial to assess cell engraftment/durability, animal models and human case reports suggest that macrophages infused via either peripheral or central veins will transiently pass through the lungs, before engrafting in the liver and spleen^{6,9}. As the primary endpoint for this trial is an improvement in MELD score, we did not plan to perform pre- and post-infusion liver biopsies that could be used to assess cell homing, we note this in the discussion (page 25, lines 13-16)

Reviewer 2:

- 1. Although the authors detailly explain the underlying disease background, the rationale for using a macrophage therapy is rather short and only briefly refers to previous studies by the group (page 9, middle paragraph). Maybe this part could be a bit extended to further balance the introduction.*

We have extended this section (page 7, lines 6-21) to provide a more nuanced rationale for our approach. Herein, we address the putative benefits of syngenic macrophages, including their antifibrotic and regenerative capacity in altering the liver niche including macrophage “switching” capability.

Additionally, we cover some of the pre-clinical evidence of homing of these cell-types to the liver parenchyma and with a proportion remaining resident in the spleen, following peripheral IV infusion. Following infusion, these cells initially by-pass the lungs to engraft in the liver, where they have been shown to preferentially sequester.^{6,9} (page 8, lines 10-17). Obviously,

these points are raised elsewhere, although this seems like an ideal section of the manuscript to cover this.

2. *It is not clear to us, why the authors decided to change the trial design regarding the numbers of infusion (accepted by the trial steering committee): Are the authors expecting the same efficacy with one infusion in comparison to three? Have there be any preclinical studies comparing the efficiencies? Why have 3 patients been treated with 3 doses? And are those also the patients on which the additional safety analysis will be performed?*

Initially, the proposed trial was designed to administer 3 infusions to those randomised to the treatment arm. It became apparent that it would not be acceptable or feasible to continue with 3 infusions due to the onerous commitment required of participants and the challenge to complete the trial within the proposed timeframe. Therefore, as a pragmatic approach that was approved by the TSC, and in line with the phase 1 study, it was decided that a single infusion protocol should be adopted to simplify the participant journey and ensure adequate recruitment. There will be a subgroup analysis of those who received the 3 infusions, prior to the protocol being amended. There is no strong pre-clinical evidence comparing the relative efficacy of single versus multiple dosing. We have adjusted the trial design section to reflect these points (page 9, lines 9-14).

3. *More from a scientific perspective: Do you think that through infusion, the majority of the macrophages will reach the target organ? Have you thought about a different cell delivery system in order to deliver macrophages directly to the damaged liver?*

These are important points to consider. Pre-clinical models of peripheral administration of macrophages have demonstrated rapid macrophage engraftment in the liver, with some cell sequestration in the spleen, having rapidly transited via the lung. Given that the phase I study⁸ was designed to assess safety and feasibility of this peripheral i.v. approach, we felt that it was justified to continue with this approach given the promising preliminary data suggesting positive effects. While as suggested there are alternative routes to be considered, including administration via portal vein, intrasplenic and hepatic artery injection, there are safety concerns and practical issues relating to each of these. These include risks of thrombosis, vascular injury¹⁰ and bleeding, splenic sequestration and presence of porto-systemic shunting/reversal of flow associated with portal hypertension.¹¹ We have covered these points in both the introduction and discussion section (page 8, lines 10-17 and page 25, lines 11-19).

4. *How was the minimum cell concentration of 1.25×10^8 defined?*

The minimum cell concentration was included as a pre-requisite that the product has a specified minimum concentration at release as stipulated by the MHRA. The cell concentration was derived from empirical models which included cell stability and viability indices. To date, no patients have not achieved the requisite cell yield. (page 14, lines 13-14)

5. *In the discussion part, you mentioned that if the treatment is effective, autologous macrophage cell therapy could improve clinical outcomes and enhance HRQoL in people with cirrhosis. Don't you want to say that if effective, the therapy will improve the outcome? Otherwise, how can you determine if the treatment is effective if it might not improve the outcome?*

We appreciate this suggestion and have reworded the text accordingly (page 25, lines 22-23). There is however also the consideration that patients may exhibit improvements in

subjective HRQoL scores within the trial, even in the event of no demonstrable clinical benefit (Hawthorne effect). This is obviously a separate issue relating to a patient's perception of health care-derived benefits for a condition where there are no approved treatment approaches apart from liver transplantation. This will likely be a discussion point from the trial outcomes.

6. *Can the authors briefly explain how the macrophages are generated from patients' monocytes and not only refer to the reference? Did the authors (previously) proof that the cells are the same whether infused 7 or 10 days post differentiation or what is the reason for that wide time range?*

We have included additional details in the introduction, which now also includes an overview of the manufacturing process (page 7, lines 23-26 and page 8, line 1-9). This process generally encapsulates the standard 7-day manufacturing process, with a 24hrs hold time before starting cell-selection and 24hrs window within to infuse the cells following QC and cell release. The validation process does include the timeframe of 7-10 days and was undertaken by our collaborators in the Scottish National Blood Transfusion Service during initial clinical phases and the release characteristics of the product take this into consideration.

Minor comments:

1. *Page 9, line 18/19 still contains a comment "REF-use one of any reviews" instead of citing a review*

We have corrected this and included the reference.

2. *Page 15, line 34/35: "as previously described24s": delete the "s" after the reference*

We have removed the "s" as suggested.

3. *Page 26, line 11/12. Please define SUSARs at the first mentioning.*

We have now defined SUSAR completely before using the abbreviation.

References

1. King A, Barton D, Beard HA, et al. REpeated AutoLogous Infusions of STem cells In Cirrhosis (REALISTIC): a multicentre, phase II, open-label, randomised controlled trial of repeated autologous infusions of granulocyte colony-stimulating factor (GCSF) mobilised CD133+ bone marrow stem cells in patients with cirrhosis. A study protocol for a randomised controlled trial. [cited 2020 Feb 24]; Available from: <http://bmjopen.bmj.com/>
2. Boulter L, Govaere O, Bird TG, et al. Macrophage-derived Wnt opposes Notch signaling to specify hepatic progenitor cell fate in chronic liver disease. *Nature Medicine* 2012;18(4):572–9.
3. Bird TG, Lu WY, Boulter L, et al. Bone marrow injection stimulates hepatic ductular reactions in the absence of injury via macrophage-mediated TWEAK signaling. *Proceedings of the National Academy of Sciences of the United States of America* [Internet] 2013 [cited 2021 Feb 22];110(16):6542–7. Available from: www.pnas.org/cgi/doi/10.1073/pnas.1302168110

4. Duffield JS, Forbes SJ, Constandinou CM, et al. Selective depletion of macrophages reveals distinct, opposing roles during liver injury and repair. *Journal of Clinical Investigation* 2005;115(1):56–65.
5. Ramachandran P, Pellicoro A, Vernon MA, et al. Differential Ly-6C expression identifies the recruited macrophage phenotype, which orchestrates the regression of murine liver fibrosis. *Proceedings of the National Academy of Sciences [Internet]* 2012 [cited 2021 Sep 2];109(46):E3186–95. Available from: <https://www.pnas.org/content/109/46/E3186>
6. Thomas JA, Pope C, Wojtacha D, et al. Macrophage therapy for murine liver fibrosis recruits host effector cells improving fibrosis, regeneration, and function. *Hepatology [Internet]* 2011 [cited 2021 Feb 22];53(6):2003–15. Available from: <https://pubmed.ncbi.nlm.nih.gov/21433043/>
7. Moore JK, Mackinnon AC, Wojtacha D, et al. Phenotypic and functional characterization of macrophages with therapeutic potential generated from human cirrhotic monocytes in a cohort study. *Cytotherapy* 2015;17(11):1604–16.
8. Moroni F, Dwyer BJ, Graham C, et al. Safety profile of autologous macrophage therapy for liver cirrhosis. *Nature Medicine [Internet]* 2019 [cited 2021 Feb 10];25(10):1560–5. Available from: <https://doi.org/10.1038/s41591-019-0599-8>
9. P SL, L C, N A, et al. Alternatively activated macrophages promote resolution of necrosis following acute liver injury. *Journal of hepatology [Internet]* 2020 [cited 2021 Sep 8];73(2):349–60. Available from: <https://pubmed.ncbi.nlm.nih.gov/32169610/>
10. BG C, RC G, LM da F, et al. Bone marrow mononuclear cell therapy for patients with cirrhosis: a Phase 1 study. *Liver international : official journal of the International Association for the Study of the Liver [Internet]* 2011 [cited 2021 Sep 19];31(3):391–400. Available from: <https://pubmed.ncbi.nlm.nih.gov/21281433/>
11. Dwyer BJ, Macmillan MT, Brennan PN, Forbes SJ. Cell therapy for advanced liver diseases: Repair or rebuild. *Journal of Hepatology* 2020;

VERSION 2 – REVIEW

REVIEWER	Sharma, Mithun Department of Gastroenterology and Hepatology, Asian Institute of Gastroenterology
REVIEW RETURNED	20-Oct-2021

GENERAL COMMENTS	Very good and well designed study . However the long term Outcomes at 6 months would be more interesting . This can at best be a bridge therapy
---

REVIEWER	Lachmann, Nico Hannover Medical School
REVIEW RETURNED	29-Sep-2021

GENERAL COMMENTS	The authors have adequately addressed all of my comments.
---